# Urinary Volatile Organic Compound Testing in Fast-Track Patients with Suspected Colorectal Cancer

**DOI:** 10.3390/cancers14092127

**Published:** 2022-04-24

**Authors:** Caroline E. Boulind, Oliver Gould, Ben de Lacy Costello, Joanna Allison, Paul White, Paul Ewings, Alfian N. Wicaksono, Nathan J. Curtis, Anne Pullyblank, David Jayne, James A. Covington, Norman Ratcliffe, Claire Turner, Nader K. Francis

**Affiliations:** 1Department of General Surgery, Yeovil District Hospital NHS Foundation Trust, Higher Kingston, Yeovil BA21 4AT, UK; caroline.boulind@nhs.net (C.E.B.); joanna.allison@ydh.nhs.uk (J.A.); nathan.curtis@ydh.nhs.uk (N.J.C.); 2Institute of Bio-Sensing Technology, Frenchay Campus, University of the West of England, Coldharbour Lane, Bristol BS16 1QY, UK; oliver.gould@uwe.ac.uk (O.G.); ben.delacycostello@uwe.ac.uk (B.d.L.C.); paul.white@uwe.ac.uk (P.W.); norman.ratcliffe@uwe.ac.uk (N.R.); 3Southwest NIHR Research Design Service, Somerset NHS Foundation Trust, Parkfield Drive, Taunton TA1 5DA, UK; paul.ewings@somersetft.nhs.uk; 4School of Engineering, University of Warwick, Coventry CV4 7AL, UK; a.wicaksono@warwick.ac.uk (A.N.W.); j.a.covington@warwick.ac.uk (J.A.C.); 5Department of Surgery, North Bristol NHS Foundation Trust, Southmead Road, Bristol BS10 5NB, UK; anne.pullyblank@nbt.nhs.uk; 6The John Goligher Colorectal Surgery Unit, St James’s University Hospital, Leeds Teaching Hospitals NHS Trust, Leeds LS9 7TF, UK; d.g.jayne@leeds.ac.uk; 7St James’s University Hospital, University of Leeds, 7 Clinical Sciences Building, Leeds LS9 7TF, UK; 8College of Health, Medicine and Life Scienes, Brunel University, Kingston Lane, Uxbridge, Middlesex UB8 3PH, UK; claire.turner@brunel.ac.uk; 9Division of Surgery and Interventional Science, University College London, London NW3 2PF, UK

**Keywords:** volatile organic compounds, colorectal cancer, fast track

## Abstract

**Simple Summary:**

The current pathway for the investigation of possible colorectal cancer includes the use of colonoscopy. This is an invasive and unpleasant procedure, and currently, a large number of those performed are normal. Previous research has demonstrated that urinary volatile organic compounds (VOCs) can be used to detect cancer, including colorectal cancer. However, these studies have only taken place in patients already known to have cancer. This study aimed to assess the role of urinary VOC analysis in the NHS two weeks wait for cancer pathway. Three analytical techniques were used to analyze urine samples of 558 patients during the standard NHS assessment pathway. It demonstrated that gas chromatography-mass spectrometry (GCMS) has excellent sensitivity and specificity for the identification of cancer and polyps in this patient group. These results show a potential role for urinary VOC analysis in the NHS cancer screening pathway, to reduce the need for invasive colonoscopy testing.

**Abstract:**

Colorectal symptoms are common but only infrequently represent serious pathology, including colorectal cancer (CRC). A large number of invasive tests are presently performed for reassurance. We investigated the feasibility of urinary volatile organic compound (VOC) testing as a potential triage tool in patients fast-tracked for assessment for possible CRC. A prospective, multi-center, observational feasibility study was performed across three sites. Patients referred to NHS fast-track pathways for potential CRC provided a urine sample that underwent Gas Chromatography-Mass Spectrometry (GC-MS), Field Asymmetric Ion Mobility Spectrometry (FAIMS), and Selected Ion Flow Tube Mass Spectrometry (SIFT-MS) analysis. Patients underwent colonoscopy and/or CT colonography and were grouped as either CRC, adenomatous polyp(s), or controls to explore the diagnostic accuracy of VOC output data supported by an artificial neural network (ANN) model. 558 patients participated with 23 (4%) CRC diagnosed. 59% of colonoscopies and 86% of CT colonographies showed no abnormalities. Urinary VOC testing was feasible, acceptable to patients, and applicable within the clinical fast track pathway. GC-MS showed the highest clinical utility for CRC and polyp detection vs. controls (sensitivity = 0.878, specificity = 0.882, AUROC = 0.896) but it is labour intensive. Urinary VOC testing and analysis are feasible within NHS fast-track CRC pathways. Clinically meaningful differences between patients with cancer, polyps, or no pathology were identified suggesting VOC analysis may have future utility as a triage tool.

## 1. Introduction

Colorectal symptoms are common but poor predictors of underlying malignancy making decisions on which patients to reassure or further investigate a frequent clinical challenge [1,2,3]. The UK established fast-track referral pathways to provide rapid assessment and investigation of suspected colorectal cancer (CRC) [4]. An ongoing increase in referrals resulted in a disproportionate decrease in cancer detection, with CRC diagnosis rates around 3–6% [5,6,7]. Disappointingly, no improvement in time to treatment nor survival has been identified [8].

The gold standard investigation for suspected CRC is colonoscopy. However, this invasive procedure requires bowel preparation, is unpleasant for patients, and carries a small risk of serious complications. The fast-track pathway requires significant resources contributing to the increasing demand for UK endoscopy services, which now deliver over 618,000 colonoscopies annually at a direct cost of £389 million [9]. Fast-track service resource requirements also risk delaying care for those with non-CRC indications for diagnostic or therapeutic endoscopy. Overall, as the vast majority of fast-track patients receive reassurance only, it can be considered that there is currently an unmet clinical need for better risk stratification for those with colorectal symptoms. 

Interest and utilization of non-invasive testing are increasing for the early detection of cancer and gastrointestinal pathology. Fecal immunochemical testing (FIT) is used within both the asymptomatic screening population and the NHS fast-track program, but an insufficient uptake rate of 34–71% is reported [10,11,12]. This raises the need for research into alternative methods that are more acceptable to patients. 

Volatile organic compounds (VOCs) have been studied as non-invasive diagnostic tools for gastrointestinal conditions including CRC [13,14,15,16,17,18,19,20,21]. Volatile organic compounds are products of human and microbiota cellular metabolism present and detectable in breath, urinary, and fecal samples. Two hundred and seventy-nine VOCs have been identified in urine [22]. Recent reviews assessing the utility of urinary VOCs in cancer detection have confirmed the ability of these VOCs to both identify several cancers and monitor progress over time [23,24,25]. The use of urinary VOCs is favored due to the non-invasive nature of urine sample collection, and the ease of collection in large volumes. These factors make urinary VOC analysis especially attractive as a screening method. More expensive and invasive tests might then be performed in a smaller subset of patients identified to have abnormal VOC profiles [23]. Several VOCs have been identified as important in the identification of CRC. These include alcohols, ketones, and aromatic compounds as well as enol ether and organosulfur compounds. Some of these identified VOCs are found at increased concentrations in CRC, and others at reduced concentrations, and the VOC profile has been found to return to normal following curative surgical treatment [25,26]. The metabolomic derangement from CRC gives variable sensitivity (30–94%) and specificity (60–94%) [27].

Although most of the research investigating urinary VOCs in the context of cancer was inspired by a need to identify screening tests, most compared samples from patients known to have CRC with samples from normal control subjects [23]. The application of VOC testing within NHS clinical pathways for CRC diagnosis has not previously been investigated in a multi-centre study. This is an important gap in the current literature because the ultimate use of urinary VOC testing for CRC must be feasible within the structure of NHS cancer diagnosis pathways. These studies, therefore, form a foundation on which to perform further research within clinical pathways such as the NHS fast-track cancer scheme.

This study aimed to test the feasibility and patient and user acceptability, of urinary VOC testing within the CRC fast-track referral pathway. Additionally, it aimed to provide initial sensitivity and specificity of urinary volatile organic compound analysis for the detection of colorectal cancer and adenomatous polyps.

## 2. Materials and Methods

A prospective, multi-center observational feasibility study was performed between August 2018 and December 2020 at three NHS Trusts (Yeovil, North Bristol and St James, Leeds). The study was granted NHS research ethics board approval (ref: 18/LO/1005) and funded by the NHIR Research for Patient Benefit (RfPB) stream (PB-PG-0416-20022). This report follows Standards for Reporting of Diagnostic Accuracy Studies (STARD) guidelines.

Patient inclusion criteria were age ≥18, referred on the fast-track pathway for suspected CRC, considered to require and be fit for colonoscopy or computed tomography (CT) colonography by the responsible clinical team, and able to provide written informed consent. Exclusion criteria included; current urinary tract infection (diagnosed by a General Practitioner (GP) or other clinicians, or urine dipstick positive for nitrites at screening); antibiotic use for any reason in the preceding 14 days; any contraindication for colonoscopy or CT colonography; any other proven or suspected cancer (excluding non-melanoma skin cancer); renal replacement therapy; the presence of an ileal conduit; indwelling urinary catheter or inability to provide a urine sample. Those enrolled were required to provide a urine sample prior to their colonic investigation(s). There were no other changes to patient care throughout the fast-track clinical pathway, which remained entirely at the discretion of their GP and local clinical teams. Colonoscopy/CT colonography findings and histological diagnoses were prospectively captured from patient records and multi-disciplinary team notes. VOC data were not provided to the responsible clinical teams.

Study design, particularly patient identification, nature and timing of urine collection, and patient questionnaire, were developed with a patient representative and the Yeovil “colon-aid” support group all of whom had previously undergone colorectal resection. The representatives identified urinary testing as the most acceptable to them, and was, therefore, adopted. Participants were asked about their experience and acceptability of providing a urine sample within the fast-track pathway. 

Patients were screened from consecutive fast-track CRC referrals and approached when they attended for colonoscopy or clinic review depending on local practice. An information sheet was provided with the opportunity to ask questions. After the provision of written informed consent patients were asked to provide a single urine sample of at least 16 mL to allow division into four aliquots. One was used for immediate urine dipstick testing (Siemens Multistix 10SG™ - Siemens [Erlangen, Bavaria, Germany]) for nitrites to assess for urinary tract infection. At this time patients also completed a short questionnaire capturing demographic and lifestyle data including smoking history, family history of bowel cancer, and current bowel symptoms. Patients who declined participation were asked, voluntarily, to anonymously complete a questionnaire about their reasons for not taking part to improve acceptance in future patients. Urine specimens were stored at −80 °C within two hours and transported to the University of the West of England for laboratory analysis. 

The three aliquots were tested using three different VOC analytical techniques: Gas Chromatography-Mass Spectrometry (GC-MS), Field Asymmetric Ion Mobility Spectrometry (FAIMS), and Selected Ion Flow Tube Mass Spectrometry (SIFT-MS). GC-MS is considered the gold standard for VOC analysis. However, it is labor-intensive and time-consuming, and as a result, both FAIMS and SIFT-MS have found favor in clinical studies [28,29,30,31]. The purpose of using three approaches was twofold. (i) to maximize the chemical information being collected, undertake a comparison of these methods, and investigate which technique(s) may offer future clinical utility; and (ii) to assess the feasibility and acceptability of each testing method within the fast-track pathway. Each testing method has its strengths and weaknesses (and there are no prior reports on the best analytic method for the triage of fast-track patients. Specific information on the instruments, setup, and techniques used is provided in Appendix A.

As this was a feasibility study, no formal sample size calculation was performed. Based on an internal audit of fast-track referrals and the likelihood of identifying cancer and or polyps, we aimed to recruit a total of 600 participants to capture around 100 CRC or polyp diagnoses. An attrition rate of 5% was added. As previously reported by our group, the sensitivity of urinary VOC testing for CRC was 80% [32]; should a similar sensitivity be observed, the 95%CI would be 71–87%. For specificity, the number of participants without CRC will be larger providing correspondingly greater precision. Areas under the receiver operating characteristic curve (AUROC) were calculated.

The study incorporated the following feasibility endpoints: recruitment rate, incorporating urine VOC analysis into the NHS fast-track pathway, and acceptability of urine collection to patients. Practical steps endpoints were also captured related to urine collection; storage, transport, and time and resources required for each testing method. 

Management and interpretation of VOC output data are presented in Appendix A. GCMS data were initially analyzed using an online metabolomic/bioinformatics program called XCMS (https://xcmsonline.scripps.edu, accessed on 10 March 2022). Resultant data from XCMS, SIFT-MS, and FAIMS were managed using SPSS^®^ v26.0, and R v3.4.2. Demographic data were reported descriptively with mean and ranges unless otherwise stated. Outcomes of clinical investigations and histological data were also reported descriptively. Patient and VOC data were analyzed in the following clinically relevant groups: (i) cancer and polyp samples against controls (ii) cancer samples against non-cancer samples (iii) cancer samples against polyp samples. *p* < 0.05 was considered significant. Initial estimates of sensitivity and specificity were calculated for each testing method using conventional logistic regression and an artificial neural network (ANN).

ANN is a computational model comprising several highly interconnected processing elements (neurons) working in unison. Neural networks process information in a similar way to the human brain allowing ANNs to improve with increasing data input. Detailed ANN descriptions are available [33,34,35]. An ANN was created with the various VOC collected by the three testing methods. An input layer, one hidden layer, and an output layer design were adopted (Figure 1). 70% of the cohort was randomly selected for ANN training, with the remainder used for testing. GC-MS and SIFT-MS data were square-root transformed to minimize the impact of any large outliers in model development. Only those variables significant at the α = 0.1 level were candidates for inclusion. ANN modeling was performed 20 times to ensure robust modeling was performed and not prone to overly capitalizing on chance idiosyncratic sample features. Receiver operator characteristics curves (ROC), AUROC, gain and lift charts, and comparison with logistic regression modeling was used for specific cross-validation of the ANN.

## 3. Results

In the 13-month recruitment period, 1714 patients were screened with 248 (14.5%) ineligible, mainly due to previous cancer (*n* = 108, Figure 2). 768 patients were approached with 558 (73%) recruited. The main reasons for not enrolling were the inability to provide an adequate urine sample (*n* = 121) or declining to consent (*n* = 49). Thirteen urine samples contained nitrites. This study had no negative impact on the fast-track pathway timings, patient care, or nature of investigations and subsequent treatment.

Patient demographic data are illustrated in Table 1. The average participant age was 64 (range 18–89) and 43.3% were female. The mean BMI was 27.2 kg/m^2^ (15.5–46.2). The majority of patients were non-smokers (87.3%), though 52% of this group had been smokers previously. There was a family history of bowel cancer in 22% of participants. Most patients reported more than one presenting symptom (Table 1), the most common being diarrhoea (*n* = 322), constipation (*n* = 294), pain (*n* = 225) and rectal bleeding (*n* = 193).

All participants stated providing a urine sample was acceptable to them whilst undergoing investigation. All but four stated they would be happy to supply a urine sample to their GP. Twenty patients who declined to take part in this research gave anonymous feedback. Reasons were evenly distributed between “too much to think about today” (*n* = 6), unable to pass urine (*n* = 5), didn’t want to provide urine (*n* = 5), and being too anxious about possible cancer diagnosis (*n* = 4).

### 3.1. Sample Testing

All samples were successfully captured, stored, and transferred to the laboratory. The impact of the COVID-19 pandemic on UK healthcare and national lockdowns caused the closure of the laboratory limiting the full analysis of all captured samples. The study management team and sponsor decided to prioritize cancer and polyp sample analysis with a representative control group.

Clinical diagnostic testing performed is summarised in Table 2. Four hundred and sixty-four participants underwent colonoscopy, of which 59% were normal. Polyps were seen in 134 (29%) and CRC seen in 18 (4%) patients. Forty colonoscopies were reported as incomplete leading to two repeats and 29 CT colonographies, with seven patients having no further investigation. In total 117 CT colonography studies were performed of which 100 (86%) were reported as normal and five (4%) suspected CRC. The remainder reported a range of non-cancer pathology including diverticular disease and inflammatory bowel disease.

Two hundred and forty-three participants had tissue sent for histopathological analysis at the time of colonoscopy. Of these, 115 were normal, 86 (15%) patients had adenomatous polyp(s), 23 (4%) had cancer (22 adenocarcinomas, one neuroendocrine tumor) and 19 had hyperplastic polyps only. Of the polyp patients, 78 were reported as low-grade dysplasia with 8 patients having a high-grade dysplastic polyp. Four CRC patients did not proceed to surgery due to advanced disease meaning there was 18 tissue confirmed CRC diagnoses with matched urine samples.

### 3.2. SIFT-MS Analysis

All analysis steps were performed manually including defrosting of samples, pre-concentration, and incubation. Approximately 12 samples could be tested per day. A total of 368 patients samples underwent SIFT-MS analysis including 18 CRC, 86 polyps, and 263 controls with 399 ions detected in total. Three of the 399 were significantly associated with cancer and a further 11 were negatively associated with cancer. The neural network model combined eight VOCs, giving good discriminatory power between cancer and non-cancer cases (sensitivity = 0.778, specificity = 0.780, AUROC = 0.872, Table 3). Altering the number of volatiles did not improve diagnostic accuracy.

Nineteen of the 399 ions were positively associated with the cancer and polyp group. A further 69 were significantly associated with the no pathology control group. The ANN failed to discriminate between cancer and polyps from controls. A six VOC model gave sensitivity = 0.6, specificity = 0.605, AUROC = 0.662.

Eleven of the 399 ions were associated with cancer samples compared with polyp samples. While a different four were significantly associated with polyps. In ANN modeling, a total of six markers provided good discriminatory powers for separating CRC from polyps (sensitivity = 0.722, specificity = 0.759, AUROC = 0.813).

### 3.3. FAIMS Analysis

All analysis steps were performed manually including defrosting of samples, pre-concentration, and incubation. Approximately 12 samples could be tested per day. Samples from 373 patients underwent FAIMS analysis including 18 cancers, 88 polyps, and 268 controls. From the output, 50 data points were selected that held discriminatory information. Due to the nature of the FAIMS measurement process, it is likely that these data points came from a smaller number of VOCs. The analysis of these data points using a random forest classifier provided good discriminatory power to differentiate cancer from non-cancer cases (sensitivity = 0.899, specificity = 0.778, AUROC = 0.855). Altering the number of data points used did not significantly alter diagnostic accuracy. 

Comparing cancer and polyps using the same number of data points, the ANN gave reasonable discrimination (sensitivity = 0.722, specificity = 0.889, AUROC = 0.855). FAIMS did not accurately differentiate the cancer and polyp group from controls (sensitivity = 0.429, specificity = 0.872, AUROC = 0.664).

### 3.4. GC-MS Analysis

All analysis steps were performed manually including defrosting of samples, pre-concentration, and incubation. Approximately 3–5 samples could be tested per day. GC-MS testing was performed on 83 patients’ samples, including 18 cancers, 31 polyps, and 34 controls. Sixty-four VOCs were detected, with 31 significantly associated with the presence of CRC. ANN modeling using six VOC-associated mass fragments (VOCMF) in combination gave excellent discriminatory power between cancer and non-cancer cases (sensitivity = 0.833, specificity = 0.815, AUROC = 0.913). The annotations of the specific biomarkers are provided in Appendix A. Appendix A lists the annotated VOCs that correspond to the significant ions identified by XCMS that differ in the cancer group vs. polyps and controls. The use of XCMS metabolomics software makes GC-MS more feasible as it allows the faster analysis of differences between groups of chromatograms. The annotated VOCs are all increased in the cancer group. Appendix A references other studies [13,17,20,24,25,36,37,38,39]. That have annotated eight of the VOCs in urine samples when analyzing a variety of cancers including CRC. Acetone and phenol were previously seen to increase colorectal cancer in agreement with our study. Dimethyldisulphide was seen to decrease in CRC in contrast to our study. Our study annotated 4 VOCs, benzenethiol, biphenyl, 1,6-dichloro-1,5-cyclooctadiene and dibenzofuran that had not previously been found in studies of cancer. Appendix A show the experimental mass spectra (incorporating the ions identified by XCMS at the relevant retention time) and the library spectra obtained from the NIST mass spectral database. Appendix A show total ion chromatograms of a cancer sample, polyp sample, and control sample respectively. 

Twenty-one of the 64 VOCMFs were significantly associated with cancer and polyps group compared to controls. Applying the neural network model, eight VOCMF gave good discriminatory power (sensitivity = 0.878, specificity = 0.882, AUROC = 0.896). Again, altering the number of VOCMF did not improve diagnostic accuracy. 

Comparing cancer samples against polyp samples, 32 VOCMF were positively associated with CRC. Using eight VOCMF the ANN gave good discrimination (sensitivity = 0.889, specificity = 0.871, and AUROC = 0.896, Table 3).

## 4. Discussion

Non-invasive testing offers potential advantages in population assessment and triage of those presenting with colorectal symptoms. Currently, the vast majority of these patients routinely receive bowel preparation and a colonoscopy to gain reassurance that they do not have CRC. In this large prospective study, we showed that urinary VOC collection and analysis were feasible within the NHS fast-track CRC pathway and acceptable to patients. Initial VOC data shows clinically meaningful differences between CRC and non-CRC patients, with sufficient diagnostic accuracy to potentially permit future clinical use. This includes assisting decision-making processes regarding which patients require further testing and who may be safely reassured. 

Our results suggest GC-MS analysis of urinary VOC offers the best sensitivity and specificity for differentiating colorectal cancer and polyps from control samples. The optimal use of VOC data as well as the exact number, nature, and threshold levels are yet to be established. Arguably, high specificity would be of most value with the eventual goal being to safely reassure patients with low-risk colorectal symptoms in primary care settings. Even a partial reduction in colonoscopy referrals would represent a meaningful improvement for patients as well as healthcare providers. A potential to redistribute resources and capacity to other patient groups could result. Although some point of care and automated VOC testing equipment is commercially available allowing high throughputs, the infrastructure is not presently in place to allow rapid GC-MS testing in large-scale populations. However, the GC-MS run time of one hour per sample encountered in this study supports clinical utility.

Presently VOC science does not allow an exact “signature” nor concentrations to be definitively stated. Even though we were able to identify specific biomarkers that are associated with CRC, it is unlikely any one VOC biomarker will display sufficient accuracy for clinical use in CRC or any other aspect of gastrointestinal clinical practice. Further research will be required to isolate the battery of the most sensitive and specific biomarkers. Additionally, whilst this was a multicentre project, the wider generalisability of our VOC data is not known and also requires a formal investigation. 

Encouragingly we report AUROC around 0.9 for clinically relevant pathology differences comparable to available reported FIT data [40]. Shaped by our patient representatives, our design resulted in superior patient uptake and acceptability data relative to fecal VOC and FIT sampling. Urinary VOC could potentially improve testing uptake improvement in those unwilling to provide fecal samples. A future study combining urinary VOC and FIT testing is considered exciting as the techniques may prove complementary and offer improved diagnostic accuracy. 

Although we successfully delivered this prospective, multi-center research within fast-track timelines, our work contains some limitations. As would be expected from a real-world cohort of this size, only 23 colorectal cancers were identified. Although this study was not powered to make formal comparisons between VOC testing modalities, it highlights the potential application of VOC testing to fulfill the clear need for non-invasive triage. Additionally, we only analyzed 83 samples with GC-MS but 400 samples using SIFT-MS and FAIMS. This may cause questions about the comparison. The small number of samples analyzed on GC-MS was due to the time taken for each sample analysis. In this study, all analysis steps were performed manually including defrosting of samples, pre-concentration, and incubation. Approximately 12 samples could be tested per day for both SIFT-MS and FAIMS, but only 3–5 per day for GCMs. The use of auto-samplers in future studies (and within clinical pathways) would increase throughput on GCMS very significantly. This was, in itself, an important feasibility outcome, and will inform the application of these testing methods within the fast-track pathway. Furthermore, the substantial difference in the numbers tested across the different methods was limited largely to the control group; we included all cancer cases and 30 polyps in the GC-MS analysis. This approach helps preserve the internal validity of the comparison. Larger studies are required to improve the understanding and modeling power of each group to provide a definitive comparison.

It is regrettable that the pandemic directly limited our ability to analyze all collected samples, particularly GC-MS testing. Theoretically, patients in this study may have undiagnosed conditions including non-CRC neoplasia, and clinical tests (including biopsy) have an acknowledged miss rate that may have altered VOC results and patient grouping. The impact of bowel preparation and potentially associated dehydration on urinary VOC levels is unknown. In keeping with the goal of the fast-track pathway, we grouped patients based on neoplastic data which risks oversimplifying the breadth of pathology that can present via fast-track routes. 

## 5. Conclusions

In conclusion, urinary VOC analysis is acceptable to patients and was successfully performed within NHS fast-track CRC pathways, identifying clinically meaningful differences in those with cancer, polyps, or no pathology. This route of non-invasive testing may have future utility as a triage tool to reduce the need for invasive testing in those presenting with colorectal symptoms.

## Figures and Tables

**Figure 1 cancers-14-02127-f001:**
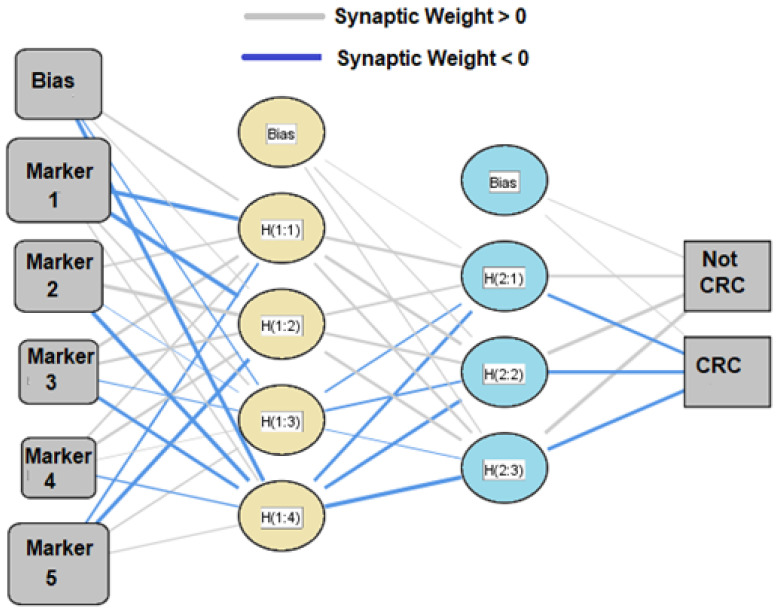
An illustrative example of an artificial neural network model consists of three layers: inputs, hidden, and output. The input represents raw information fed into the network.

**Figure 2 cancers-14-02127-f002:**
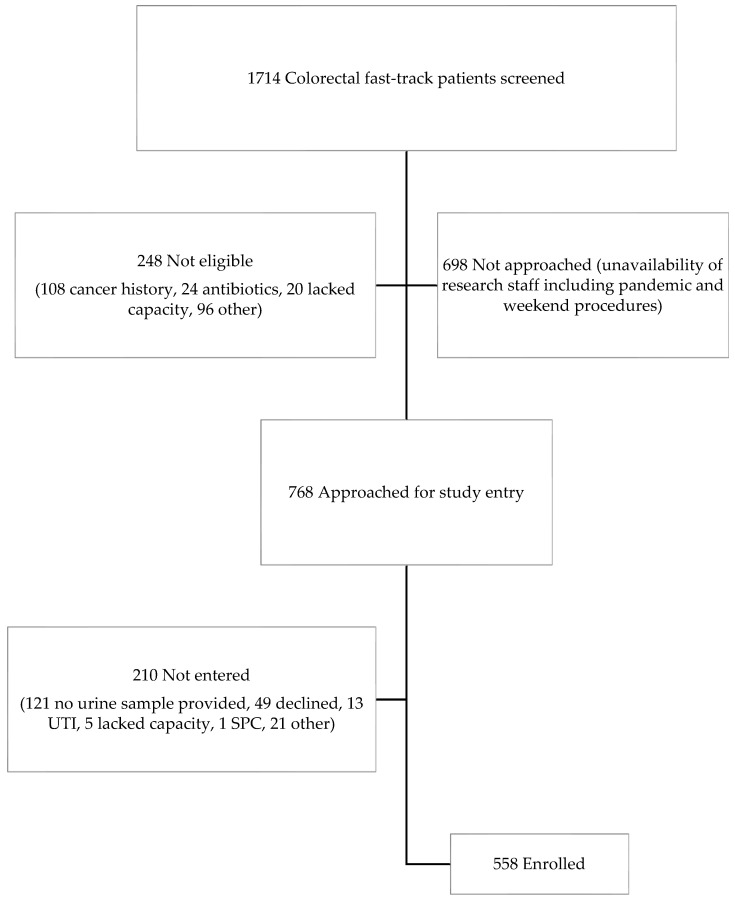
PRISMA participant flow chart for study participants.

**Table 1 cancers-14-02127-t001:** Demographic data of participants enrolled in the study. Data are means (range) or number of patients.

Demographic Detail		Result
Age (years)		64 (18–89)
Sex (female)		247 (44.3%)
Weight (kg)		78.7 (41.3–144)
Height (cm)		169.4 (121–195)
Body mass index (kg/m^2^)		27.2 (15.5–46.2)
Smoking status		
	Current	68 (12.2%)
	Past	256 (45.9%)
	Never	234 (41.9%)
Family history of colorectal cancer		124 (22.2%)
Patient-reported symptoms		
	Diarrhoea	322 (57.7%)
	Constipation	294 (52.7%)
	Pain	225 (40.3%)
	Rectal bleeding	193 (34.6%)
	Weight loss	123 (22.0%)
	Loss of appetite	98 (17.6%)

**Table 2 cancers-14-02127-t002:** Clinical tests and summary findings were performed on enrolled patients. CTC–CT colonography. CRC—colorectal cancer. Data are numbers of patients.

Outcome		Colonoscopy (*n* = 464)	CT Colonography (*n* = 117)
Normal		272 (58.6%)	100 (85.5%)
Abnormal		152 (32.8%)	17 (14.5%)
	Polyp	134 (28.9%)	
	CRC	18 (3.9%)	5 (4.3%)
Incomplete		40 (8.6%)	

**Table 3 cancers-14-02127-t003:** Diagnostic accuracy data with 95% confidence intervals for each patient group and volatile organic compound method. AUROC—area under the receiver operator curve.

Cancer vs. Non-Cancer	SIFT-MS	FAIMS	GCMS
Sensitivity	0.778 (0.524, 0.936)	0.889 (0.653, 0.986)	0.833 (0.586, 0.964)
Specificity	0.780 (0.733, 0.822)	0.778 (0.524, 0.936)	0.815 (0.700, 0.901)
AUROC	0.872 (0.794, 0.949)	0.855 (0.724, 0.986)	0.913 (0.825, 1.000)
Cancer and Polyps vs. Control			
Sensitivity	0.600 (0.500, 0.694)	0.429 (0.332, 0.529)	0.878 (0.752, 0.953)
Specificity	0.605 (0.543, 0.664)	0.872 (0.794, 0.928)	0.882 (0.726, 0.967)
AUROC	0.662 (0.602, 0.723)	0.664 (0.591, 0.734)	0.896 (0.802, 0.966)
Cancer vs. Polyps			
Sensitivity	0.722 (0.465, 0.903)	0.722 (0.465, 0.903)	0.889 (0.653, 0.986)
Specificity	0.759 (0.655, 0.844)	0.889 (0.653, 0.986)	0.871 (0.702, 0.964)
AUROC	0.813 (0.704, 0.922)	0.855 (0.732, 0.977)	0.896 (0796–0.996)

## Data Availability

Deidentified VOC output could be shared (with no end date) subject to the approval of a proposal and completion of a data-sharing agreement and/or ethical approval.

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
