# Peer review of "Urinary Volatile Organic Compound Testing in Fast-Track Patients with Suspected Colorectal Cancer"

_cancers, 2022, doi:10.3390/cancers14092127_

Round 1

Reviewer 1 Report

The manuscript by C.E. Boulind et al. is devoted to the possibility of using analysis of volatile organic compounds in urine as an additional tool for cancer detection. Undoubtedly, research topics are extremely relevant. Work in this direction contributes to the emergence of new, informative and reliable approaches to the diagnosis of oncological diseases, including colorectal cancer. The authors compared the results of urine analysis of patients obtained using three analytical methods that are widely used in clinical practice. The manuscript corresponds to the scope of the Cancers journal and can be published after some amendments related to issues that must be additionally clarified:

  1. Section «Introduction». In my opinion, the authors did not pay attention to what was previously published on the analysis of VOCs in urine for cancer diagnosis. Need specifics. You are not the first to conduct such research. What has been achieved, what limitations or disadvantages? The same section lacks information on the methods used in clinical practice for diagnosing CRC. On this basis, the choice of urinalysis methods described in the manuscript would be understandable.
  2. The authors compared 3 analytical methods for the analysis of VOCs in urine for cancer diagnosis. What new has been obtained in comparison with what is known in the literature? It's not clear from the manuscript.
  3. The authors claim that GC-MS has the highest sensitivity and specificity. However, this method was used to analyze 83 samples, while using SIFT-MS and FAIMS about 400. Is it possible to make a comparison in this case?
  4. Line 281. The authors talk about the identification of specific biomarkers. However, these components are not included in the manuscript. This information must be added.
  5. It is also desirable to provide chromatograms and mass spectra of urine samples from different groups of patients (Cancer vs non-Cancer). This information can be provided in Supplementary materials.

By its content, the work corresponds to the profile of the journal "Cancers" and can be published after corrections.

Author Response

The manuscript by C.E. Boulind et al. is devoted to the possibility of using analysis of volatile organic compounds in urine as an additional tool for cancer detection. Undoubtedly, research topics are extremely relevant. Work in this direction contributes to the emergence of new, informative and reliable approaches to the diagnosis of oncological diseases, including colorectal cancer. The authors compared the results of urine analysis of patients obtained using three analytical methods that are widely used in clinical practice. The manuscript corresponds to the scope of the Cancers journal and can be published after some amendments related to issues that must be additionally clarified:

  1. Section «Introduction». In my opinion, the authors did not pay attention to what was previously published on the analysis of VOCs in urine for cancer diagnosis. Need specifics. You are not the first to conduct such research. What has been achieved, what limitations or disadvantages? The same section lacks information on the methods used in clinical practice for diagnosing CRC. On this basis, the choice of urinalysis methods described in the manuscript would be understandable.

Response: many thanks for the reviewer’s comment.  We have now updated the literature and added a number of recent references that are related to cancer detection and urine analysis. This section has been added to the introduction and the references are updated.  The added section in the introduction states:

Volatile organic compounds (VOCs) have been studied as non-invasive diagnostic tools for gastrointestinal conditions including CRC.[13-21] Volatile organic compounds are products of human and microbiota cellular metabolism present and detectable in breath, urinary and faecal samples. Two hundred and seventy nine VOCs have been identified in urine.[22] Recent reviews assessing the utility of urinary VOCs in cancer detection have confirmed the ability of these VOCs to both identify a number of cancers, and monitor progress over time.[23-25] The use of urinary VOCs is favoured due to the non-invasive nature of urine sample collection, and the ease of collection in large volume. These factors make urinary VOC analysis especially attractive as a screening method. More expensive and invasive tests might then be performed in a smaller subset of patients identified to have abnormal VOC profiles.[23] A number of VOCs have been identified as important in the identification of CRC. These include alcohols, ketones and aromatic compounds as well as enol ether and organosulfur compounds. Some of these identified VOCs are found at increased concentrations in CRC, and others at reduced concentration, and the VOC profile has been found to return to normal following curative surgical treatment.[25, 26] The metabolomic derangement from CRC gives variable sensitivity (30%-94%) and specificity (60-94%)[27].

Although most of the research investigating urinary VOCs in the context of cancer was inspired by a need to identify screening tests, most compared samples from patients known to have CRC with samples from normal control subjects with very little report on the application of VOC to test for polyps.[23] We included polyp diagnosis in our sample in order to replicate conditions in NHS pathways. The application of VOC testing within NHS clinical pathways for CRC diagnosis has not been investigated. This is an important gap in the current literature because the ultimate use of urinary VOC testing to triage suspected cancer patients for invasive investigations such as colonoscopy  within the structure of NHS cancer diagnosis pathways. This study  therefore will inform further research within clinical pathways such as the NHS fast-track cancer pathways.”

  1. The authors compared 3 analytical methods for the analysis of VOCs in urine for cancer diagnosis. What new has been obtained in comparison with what is known in the literature? It's not clear from the manuscript.

Response: Thanks for highlighting this point.  The aim of this study was to test the feasibility of urinary VOC testing within the CRC fast-track referral pathway.  We choose three analytic methods to test the feasibility of applying each method within the fast track referral for suspected colorectal cancer as each testing method has its strength and weakness and there is no prior reports on testing all of them within the fast track pathway.  GC-MS is considered the gold standard, but it is more labour intensive and time consuming compared to FAIMS and SIFT-MIS.  We have edited the method section in page 3, to highlight this point. 

“The three aliquots were tested using three different VOC analytical techniques: Gas Chromatography Mass Spectrometry (GC-MS), Field Asymmetric Ion Mobility Spectrometry (FAIMS) and Selected Ion Flow Tube Mass Spectrometry (SIFT-MS). GC-MS is considered the gold standard for VOC analysis however it is labour intensive and time consuming, hence both FAIMS and SIFT-MS have found favour in clinical studies.[24-27] The purpose of using three approaches was two fold: (i) to maximize the chemical information being collected, undertake a comparison of these methods, and investigate which technique(s) may offer future clinical utility; and (ii) to assess feasibility and acceptability of each testing method within the fast track pathway. Each testing method has its strengths and weakness (and there are no prior reports on the best analytical method that can be used to triage fast track patients).  Specific information on the instruments, setup and techniques is provided in a supplementary materials ( appendix A).

  1. The authors claim that GC-MS has the highest sensitivity and specificity. However, this method was used to analyze 83 samples, while using SIFT-MS and FAIMS about 400. Is it possible to make a comparison in this case?

Response: we acknowledge this point as we only analysed 83 samples with GC-MS but 400 using FAIMS and SIFT-MS.  The main aim of this study was to test the feasibility of VOC testing within the clinical pathway.  We have identified that  the time taken and the throughput of analysing 83 samples with GC-MS was similar to analysing 400 samples for the other two methods.  This was in itself an important feasibility finding to inform the main aim of this study. We have updated the conclusion of the abstract acknowledging the labour intensive nature of the GC-MS. Additionally, although we only analysed 83 samples using GC-MS, the main reduction was in the number of the control samples as we included all cancer samples and 30 polyps.  This approach helps preserve the internal validity of the comparison.  We hope that our findings can inform future larger studies with equal groups to provide definitive comparison.  We have edited the manuscript to highlight these points in the discussion and limitation (page 11) to read:

Additionally, we only analysed 83 samples with GC-MS but 400 samples using SIFT-MS and FAIMS. This may cause questions about the comparison. The small number of samples analysed on GC-MS was due to the time taken for each sample analysis. In this study, all analysis steps were performed manually including defrosting of samples, pre-concentration and incubation. Approximately 12 samples could be tested per day for both SIFT-MS and FAIMS, but only 3-5 per day for GCMs. The use of auto-samplers in future studies (and within clinical pathways) would increase throughput on GCMS very significantly. This was, in itself, an important feasibility outcome, and will inform the application of these testing methods within the fast-track pathway. Furthermore, the substantial difference in the numbers tested across the different methods was limited largely to the control group; we included all cancer cases and 30 polyps in the GC-MS analysis. This approach helps preserve the internal validity of the comparison. Larger studies are required to improve understanding and modelling power of each group to provide definitive comparison.

  1. Line 281. The authors talk about the identification of specific biomarkers. However, these components are not included in the manuscript. This information must be added.

Response: We would like to thank the reviewer for raising this important point.  We have now added the specific biomarkers (appendix C).  We have also indicated which ones were previously identified in cancer detection and the new biomarkers.  We have added a paragraph in the manuscript referring to the specific biomarkers, as follows:

“Table S1. lists the annotated VOCs that correspond to the ions identified by XCMS that differ between the cancer group vs polyps and controls. The use of XCMS metabolomics software makes GC-MS more feasible as it allows the faster analysis of differences between groups of chromatograms.  The annotated VOCs are all increased in the cancer group. Table S1 references other studies that have annotated eight of the VOCs in urine samples when analysing a variety of cancers including colorectal cancer. Acetone and phenol were previously seen to increase in colorectal cancer in agreement with our study. Dimethyldisulphide was seen to decrease in CRC in contrast to our study. Our study annotated 3 VOCS, benzenethiol, biphenyl and dibenzofuran that had not previously been found in studies of cancer. Figures S1-S12 show the experimental mass spectra (incorporating the ions identified by XCMS at the relevant retention time) and the library spectra obtained from the National Institute of Standards and Technology (NIST) mass spectral database (version 2.2, 2014, Gaithersburg, MD, USA). Figures S13-S15 show total ion chromatograms of a cancer sample, polyp sample and control sample respectively.”

  1. It is also desirable to provide chromatograms and mass spectra of urine samples from different groups of patients (Cancer vs non-Cancer). This information can be provided in Supplementary materials.

Response: We have included this in the same appendix along with the annotations of the biomarkers in  appendix C.

By its content, the work corresponds to the profile of the journal "Cancers" and can be published after corrections

Reviewer 2 Report

The authors have designed a comprehensive investigation of three analytical approaches, GCMS, FAIMS and SIFT-MS to assess urinary VOC for potential non-invasive colorectal cancer diagnosis in the fast-track pathway. The outcome of the study could have a significant impact on the current colorectal cancer screening pathway, providing substantial reductions in both medical resource needs and costs. Although the listed specificity, sensitivity, etc. for all three approaches are looking very promising, especially GCMS, no annotations on the VOC were provided throughout the manuscript. Annotations on the identified VOC are essential for the evaluation of the validity of the groups of VOC that had been used in the study to differentiate patient groups. Identification/annotation of the VOC should be provided for the publication of the manuscript.

There are also 3 minor points:

  1. Fig 1, the legend description is not complete.
  2. Line 204, 117 CT colonography were performed, 100 normal and 5 suspected CRC. What about the other 12 (117-100-5)?
  3. Line 206, 243 participants had tissue sent for histopathological analysis – how was this number chosen? Please explain.

Author Response

The authors have designed a comprehensive investigation of three analytical approaches, GCMS, FAIMS and SIFT-MS to assess urinary VOC for potential non-invasive colorectal cancer diagnosis in the fast-track pathway. The outcome of the study could have a significant impact on the current colorectal cancer screening pathway, providing substantial reductions in both medical resource needs and costs. Although the listed specificity, sensitivity, etc. for all three approaches are looking very promising, especially GCMS, no annotations on the VOC were provided throughout the manuscript. Annotations on the identified VOC are essential for the evaluation of the validity of the groups of VOC that had been used in the study to differentiate patient groups. Identification/annotation of the VOC should be provided for the publication of the manuscript.

Response: This has now been addressed (in response to the first reviewer) and all information are now added as appendix C.

There are also 3 minor points:

  1. Fig 1, the legend description is not complete.

Response: Thank you, the final few words were added to the legend in error. Figure 1 legend now reads as follows

‘Figure 1: Illustrative example of an artificial neural network model consisting of three layers: inputs, hidden and output. The input represents raw information fed into the network.’

  1. Line 204, 117 CT colonography were performed, 100 normal and 5 suspected CRC. What about the other 12 (117-100-5)?

Response: Thank you for pointing out this omission. A sentence has been added to clarify this situation as follows (line 224);

‘The remainder reported a range of non-cancer pathology including diverticular disease and inflammatory bowel disease.’

  1. Line 206, 243 participants had tissue sent for histopathological analysis – how was this number chosen? Please explain

Response: This is the number of patients who had tissue sent for histology at the time of colonoscopy. This has been clarified by the addition of a couple of words (line 227);

“Two hundred and forty-three participants had tissue sent for histopathological analysis at the time of colonoscopy”

Best Regards

Nader Francis

Round 2

Reviewer 1 Report

The authors of the manuscript have made significant, in my opinion, corrections, making it more understandable for readers. Most of my comments have been taken into account. The only thing I want to pay attention to:
- Is it important to present library and experimental mass spectra (GCMS) in supplementary materials?

In general, the manuscript corresponds to the profile of the journal "Cancers" and can be accepted for publication in present form.

Author Response

Dear Editor,

We would like to thank you for your speedy and positive response.

Yes, we feel Is it necessary. We included the experimental and library mass spectra as the annotations are tentative and one of the compounds that increased with cancer in this study was unknown. Therefore, providing the experimental and library mass spectra allows other researchers to assess the quality of the annotations from this study. It also allows them to compare their experimental mass spectral data for VOCs associated with colonic cancer or other cancers with the annotations from this study. 

We have uploaded here an updated supplementary materials to replace the previous ones. 
